developmental biology/environmental science/ecology

*Drosophila suzukii*, ovipositor, plasticity, geometric morphometrics, development, invasion

**Author for correspondence:**
Raphaël Cornette
e-mail: raphael.cornette@mnhn.fr

[†] Present address: Ecological Genetics Research Unit, Organismal and Evolutionary Biology Research Programme, Faculty of Biology and Environmental Sciences, Biocenter 3, Viikinkaari 1, FIN-0014, University of Helsinki, Finland.
[‡] These authors contributed equally to this study.

# Limited thermal plasticity and geographical divergence in the ovipositor of *Drosophila suzukii*

Ceferino Varón-González[1], Antoine Fraimout[2,†], Arnaud Delapré[1], Vincent Debat[1,‡] and Raphaël Cornette[1,‡]

[1]Institut de Systématique, Evolution, Biodiversité (ISYEB), Muséum National d'Histoire Naturelle, CNRS, Sorbonne Université, EPHE, Université des Antilles, 57 rue Cuvier, CP 50, 75005 Paris, France
[2]Centre de Biologie pour la Gestion des Populations, UMR CBGP, INRA, CIRAD, IRD, Montpellier SupAgro, University of Montpellier, 755 avenue du Campus Agropolis CS 30016, 34988 Montferrier sur Lez cedex, France

 CV-G, 0000-0002-4577-4187; AF, 0000-0003-4552-3553;
VD, 0000-0003-0040-1181; RC, 0000-0003-4182-4201

Phenotypic plasticity has been repeatedly suggested to facilitate adaptation to new environmental conditions, as in invasions. Here, we investigate this possibility by focusing on the worldwide invasion of *Drosophila suzukii*: an invasive species that has rapidly colonized all continents over the last decade. This species is characterized by a highly developed ovipositor, allowing females to lay eggs through the skin of ripe fruits. Using a novel approach based on the combined use of scanning electron microscopy and photogrammetry, we quantified the ovipositor size and three-dimensional shape, contrasting invasive and native populations raised at three different developmental temperatures. We found a small but significant effect of temperature and geographical origin on the ovipositor shape, showing the occurrence of both geographical differentiation and plasticity to temperature. The shape reaction norms are in turn strikingly similar among populations, suggesting very little difference in shape plasticity among invasive and native populations, and therefore rejecting the hypothesis of a particular role for the plasticity of the ovipositor in the invasion success. Overall, the ovipositor shape seems to be a fairly robust trait, indicative of stabilizing selection. The large performance spectrum rather than the flexibility of the ovipositor would thus contribute to the success of *D. suzukii* worldwide invasion.

# 1. Introduction

Phenotypic plasticity is a pervasive feature in nature [1] and a major response to changing environmental conditions [2]. Because it may facilitate the colonization of new environments (e.g. [3]), it has been suggested that plasticity may play an important role in biological invasions: accordingly, invasive populations are expected to be more plastic than non-invasive populations [3–7]. Although often discussed theoretically [8,9], this hypothesis has been comparatively rarely tested [6], in particular in animal species [10,11].

Drosophila suzukii has received much attention over the last 10 years, as it has colonized multiple countries worldwide [12] and induced severe losses in agriculture [13–15]. This species has been extensively collected to test hypotheses about the role of plasticity during its invasion (e.g. [10,16–18]). However, plasticity largely depends on the environmental factor considered and the morphological trait under study [10,19–21]. For D. suzukii, the temperature has been frequently chosen as the factor inducing phenotypic plasticity due to its pervasive effect on insect development (e.g. [22–24]) and its importance in shaping the distribution of Drosophila species [24]. Different morphological structures such as wings, thorax and ovipositor have been investigated (e.g. [10,16,18]). The ovipositor is a particularly interesting structure owing to the reproductive behaviour of this species: D. suzukii's damaging potential is indeed due to its overdeveloped ovipositor, used to pierce through the skin of ripening fruits and lay its eggs [25]. It is well known that fruit texture is strongly affected by temperature (e.g. [26]): specifically, their firmness and resistance to puncture tends to decrease with increasing temperature (e.g. [27]). It is thus conceivable that D. suzukii ovipositor might present some adaptive plasticity to temperature, allowing it to pierce fruit skins of (thermally induced) varying resistance. An alternative hypothesis is that it might rather be under stabilizing selection, as has been suggested in Drosophila melanogaster for genitalia [21], in which case we should expect a reduced sensitivity to temperature.

The ovipositor is a microscopic three-dimensional structure (about 500 µm). Three-dimensional characterization of its shape is essential to recover all the possible features involved in its performance and therefore to link its morphology to the possible selective forces affecting it. Two-dimensional approximations of three-dimensional structures might be troublesome because all the variation recovered by one physical dimension would be missing and that might affect the analysis [28,29]. Finally, the complete description of shape may be particularly important for assessing the ovipositor plasticity: a two-dimensional analysis could lead to underestimations of the plastic shape change when the plastic variation is not recovered among the shape descriptors. We thus developed an approach based on the combination of scanning electron microscopy (SEM)-based photogrammetry and three-dimensional geometric morphometrics allowing to finely depict and quantify the ovipositor three-dimensional shape and its variation.

In this study, we analyse the plastic response of the ovipositor shape to developmental temperature in three different geographical populations of D. suzukii, including a population from the native range (Japan) and two populations from the invaded range (France and USA). These three geographical populations represent the three most genetically divergent populations of the distribution [12]. By contrasting laboratory lines derived from native and invasive populations, we (i) investigate whether there is any genetic divergence in the ovipositor shape across the distribution range; (ii) quantify the ovipositor plasticity to temperature; and (iii) investigate whether plasticity is higher in invasive populations, as predicted if plasticity played a role in the invasion success, possibly allowing D. suzukii to exploit a larger diversity of substrates in varying thermal conditions.

# 2. Material and methods

## 2.1. Samples

Adult flies were sampled in 2014 using banana bait traps and net swiping in three different regions: one belonging to the native range (Sapporo, Hokkaido, Japan) and two to the invasive range (Paris, France and Dayton, Oregon, USA). Ten isofemale lines per locality were stocked so that they performed single matings separately and the F1 offspring was expanded in consecutive series of vials [30]. These stocks were maintained at 22°C on a medium with corn starch, yeast with antibiotics and hydroxyl-4 benzoate. Female flies were left to oviposit for 24 h in two separate sets of 20 vials and after oviposition was checked parent flies were removed. Then, two batches were placed in two incubators: one set of eggs was stored at 16°C and the other one at 28°C (keeping a third at 22°C). Therefore, for each population and temperature, we produced 10 isofemale lines in separate rearing vials with single matings at three different

(*a*)

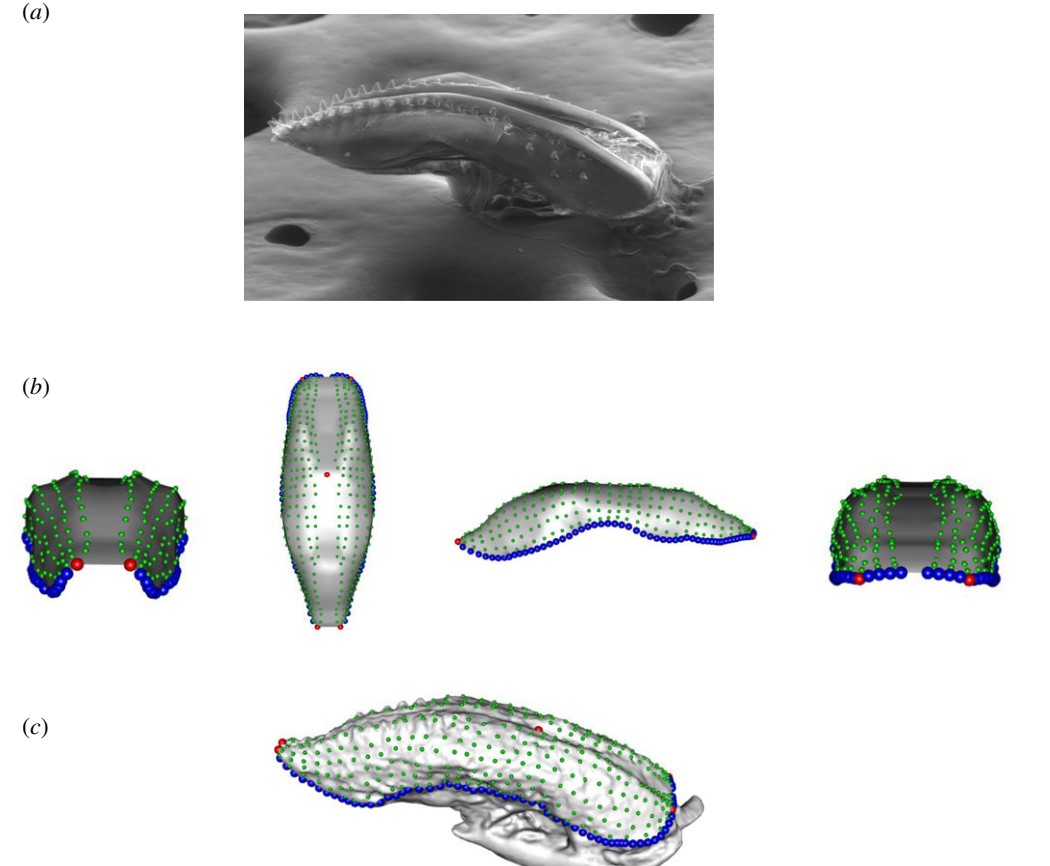

(*b*)

(*c*)

**Figure 1.** Ovipositor by electronic microscopy (*a*), template (*b*) and ovipositor phenotyping (*c*). Once the ovipositor pictures by electronic microscopy were obtained (*a*) and the three-dimensional reconstruction of the ovipositor was done, we build a template with a simplified shape of an ovipositor (*b*) where we placed landmarks (red), semilandmarks (blue) and surface semilandmarks (green). This template was then projected to each three-dimensional reconstruction to obtain the three-dimensional landmarks characterizing the ovipositor shape (*c*).

experimental temperatures: i.e. 30 lines per geographical population. The position of the incubators was assigned randomly and they were kept at the experimental temperatures until 2 days after the emergence. These experiments were originally conducted to run the analyses published in Fraimout *et al.* [10].

Final samples consisted of 20 individuals from Paris raised at 16°C, 11 at 22°C and 13 at 28°C; 19 individuals from Sapporo raised at 16°C, 20 at 22°C and 23 at 28°C; and 14 individuals from Dayton at 16°C, 6 at 22°C and 13 at 28°C.

## 2.2. Electronic microscopy

For each fly, the ovipositor was detached from the body—the two valves being kept in connection—and the connective tissues were manually removed. Because all the specimens were conserved in alcohol, no deformation was produced during the removal of the ovipositors. Then, they were photographed using an environmental scanning electron microscope (ESEM). Images were collected in low vacuum (0.37 Torr) with a large field low vacuum secondary electron detector using a FEI Quanta 200 FEG operating at 15 kV at a working distance of 10 mm.

From each ovipositor, 52 pictures were taken describing two semicircular trajectories, perpendicular between them. That allowed recovering information from all different angles of each specimen.

## 2.3. Photogrammetric reconstruction

The three-dimensional reconstruction of each ovipositor was inferred using photogrammetry (figure 1), the technique allowing the three-dimensional representation of an object from a set of pictures. The photogrammetric process starts with the alignment of the pictures obtained from the ESEM, i.e. the

recognition of analogous parts among pictures. Where difficulties for the picture alignment were found, a mask was applied to select just the ovipositor within the pictures and discard the background, facilitating the correct alignment of the pictures. The inference of the distances among analogous pixels allows the inference of the position of these pixels in a three-dimensional space (i.e. the transformation of pixels in voxels). Once this first point cloud was inferred, all the voxels not corresponding to the ovipositor itself were removed. This cleaning fastens the next step, the re-examination of the picture alignment once a first point cloud was built in order to obtain more analogous voxels. As a result, from the first point cloud, we obtained a dense cloud. Finally, a mesh was built based on the dense cloud with no *a priori* about the final shape (arbitrary surface type). All reconstructions were done in PhotoScan [31].

Because many of the reconstructions were built using a mask, the scale bar present in the pictures did not appear in the three-dimensional reconstructions and therefore we could not give the correct scale of each three-dimensional model during the reconstruction process. For that, once the three-dimensional models were obtained, we measured the real lengths of the ovipositors in the pictures using ImageJ 1.51j8 [32] and then we scale each ovipositor in MeshLab v. 2016.12 [33]. The advantage of MeshLab is that the linear measurements of the object do not consider its surface curvature (i.e. it uses Euclidean distances), the same as picture measurements. In any case, to avoid any possible deformation due to the picture perspective we used the dorsal pictures of the ovipositor and the dorsal three-dimensional view of the ovipositor (the flattest part).

## 2.4. Morphometric analyses

A set of five landmarks and three curves containing in total 130 semilandmarks were defined in each three-dimensional mesh (figure 1) [34]. One pair of landmarks was fixed at the most distal part of the ovipositor and the other pair at the most proximal part. The fifth fixed landmark was placed on the dorsal area, at the ovipositor opening. Two curves with 60 semilandmarks each were placed on the ovipositor sides. The other 10 semilandmarks surround the proximal area of the ovipositor. Landmarking was performed on Landmark Software [35]. Then, we created a template replicating a simplified form of an ovipositor (figure 1), composed of 394 surface points. Landmarks, semilandmarks and 394 surface semilandmarks were digitized on the template and they were used to deform the template via thin plate spline. Finally, all landmarks were projected on the ovipositor and they slid to minimize bending energy taking into account the ovipositor object symmetry [36]. In total, 529 landmarks described the ovipositor shape for each individual. This process follows the protocol described by Botton-Divet *et al.* [37]. The template was created with Meshlab [33] and the position of these landmarks and the subsequent sliding were performed with the R package Morpho [38].

To assess the quality of the three-dimensional shape reconstruction, we replicated the reconstruction process five times on two individuals from the same geographical population and raised at the same temperature (two Sapporo individuals raised at 16°), so the variance between individuals was minimized as much as possible. The reconstructions were done on each one of these two individuals five times and the landmarks were collected on each of the 10 meshes. A multivariate model was run with the function *procD.lm* [39] to test for the amount of variance explained by inter-individual variation in relation to the variation explained by the reconstruction and landmarking processes (residuals).

Differences among populations and temperatures were explored using a between-group principal component analysis [40]. A permutation test was run, and this transformation of the space was computed to test the significance of the differences among populations. Individuals from all groups (populations and temperature) were randomly shuffled and new pairwise Procrustes distances among group means computed. 10 000 permutations were run and significance levels obtained as the proportion of Procrustes distances less extreme. Because the permutations are run on the whole sample and significance tests are not independent, no correction for multiple comparisons is needed. The effect on the shape of the temperature and population factors as well as their interaction were tested with a linear multivariate model and permutation tests as performed in the geomorph function *procD.lm* [39]. The effect size for each factor was assessed by $Z$, an estimator based on the $F$-statistic [41]. The effect of the two factors on the centroid size was assessed with a two-way ANOVA.

To further compare the plastic responses among populations, we used the trajectory analysis method developed by Collyer & Adams [42]. This approach specifically tests the similarity between trajectories depicting shape changes in the multivariate shape space and it can be readily transposed to the analysis of shape reaction norms. With this analysis, three different aspects of the shape change are studied: the amount of shape change as the trajectory path length (size), the pattern of shape covariation as the difference in angles among the first principal component of each trajectory (direction) and the differences

in trajectory shapes (shape) as Procrustes distances between pairs of phenotypic trajectories. Although these three aspects of plasticity are somehow related, they do not look at the same effects temperature may have on shape variation. The amount of shape change reflects whether the effect of temperature on shape variation (i.e. plasticity) is larger or smaller in some populations. The pattern of shape covariation whether temperature affects the correlation patterns among the different landmarks, changing which landmarks covary together in response to temperature and in the degree and sign of these covariations. Last, the differences in trajectory shapes study the existence of differences among shapes at the same temperature. The statistical assessment of these three features is based on the simulated resampling from a distribution characterized by the difference in path length, angle or distances, respectively, and their standard deviations.

Allometry was quantified using a linear model of the logarithm of the centroid size against symmetric shape [43]. A general allometric pattern was expected given the pervasive effect of temperature on size in insects [24] as well as previously published effects in two dimensions [16]. Differences in the allometric slopes among geographical populations were also assessed. Because the allometric patterns are expected to be primarily influenced by temperature variation, we would expect the differences in allometric slopes and the differences in reaction norms to be analogous. Differences among slopes were tested with an ANOVA. All morphometric tests were applied in the R package geomorph [39].

Finally, we investigated the degree of relative robustness of the ovipositor, by comparing its variation with that of the wing, as assessed on the same samples by Fraimout *et al.* [10]. For size, we simply computed the coefficient of variation (CV) both within and among temperatures (i.e. among mean centroid sizes per temperature), for both structures. To test for differences in the CV of size between structures, we used the modified signed-likelihood ratio test, as computed in the R package *cvequality* [44]. Comparing the shape variability of two different objects is challenging, because they lie in different shape spaces and no direct multivariate extension of the CV can be applied. We used Mahalanobis distances among temperatures, computed independently for the two structures, as a measure of their relative sensitivity to temperature. Because this distance measures the difference between groups relative to the within-group variation [45], it could be comparable between structures with a note of caution. Because our hypothesis is that the ovipositor will show more robustness, it may also show lower within-group variation and that would increase distances. In this particular case, this feature makes our estimates more conservative but further applications of this method would require appropriate justification. As distance measures are affected by the data dimensionality, we estimated the Mahalanobis distances on the same number of principal components for each dataset (26 principal components: 100% of the fly shape variation and 96.94% of the ovipositor shape variation). The statistical assessment of the difference in plasticity between temperatures was done with an *ad hoc* permutation test under the null hypothesis that the difference in plasticity between structures is equal to zero. In it, for the ovipositor and the wing independently, all distances from individuals at one temperature to individuals at another temperature were estimated. Then, all within-structure among-temperature distances were randomly assigned to two groups and the difference between the means of these two groups stored. The permutation was run 1000 times and a distribution of differences in plasticity (centred in zero) was generated. The empirical differences were considered significant if their real value fell out of the 95% of the values in the distribution. To obtain the distances among temperatures, we applied the function *CVA* in the R package Morpho [38]. All analyses and data management were conducted in RStudio version 1.1.442 [46].

# 3. Results

The three-dimensional shape reconstruction of the ovipositors allowed us to assess the ovipositor three-dimensional shape variation precisely. We found a significant effect of temperature and geographical variation on the ovipositor size and three-dimensional shape, but the effects appeared weak and all nine experimental groups were not fully discriminated (table 1). Although the interaction between geographical and temperature factors was significant in the multivariate model, no differences among shape trajectories or allometric slopes in response to temperature were detected among geographical populations (figure 2).

## 3.1. Measurement error

The repeated reconstruction of the three-dimensional shape of the two individuals from Sapporo raised at 16°C showed that the variation in the reconstruction process was almost four times smaller than

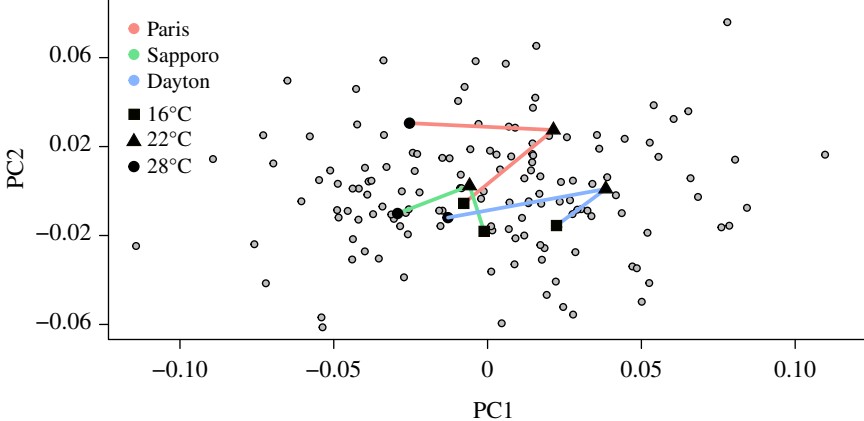

**Figure 2.** Ovipositor three-dimensional shape variability and plasticity trajectories in response to developmental temperature. First two principal components of the ovipositor shape for individuals (grey) and temperature means for each geographical population (black; square: 16°C, triangle: 22°C, circle: 28°C). The three temperature levels for each geographical population are joined so the reaction norms can be visualized for each population (Paris: red, Sapporo: green, Dayton: blue). We can observe the overlap among reaction norms and the similarity in their trajectories, suggesting similar plasticities among populations.

**Table 1.** Discriminant analysis for temperature and geographical factors. 1000 permutations using Procrustes distances between group means were run with the function *groupPCA* of the R package Morpho. Non-significant results are shaded.

|  | Paris 16° | Paris 22° | Paris 28° | Sapporo 16° | Sapporo 22° | Sapporo 28° | Dayton 16° | Dayton 22° |
|---|---|---|---|---|---|---|---|---|
| Paris 22° | 0.0058 |  |  |  |  |  |  |  |
| Paris 28° | 0.0082 | 0.0104 |  |  |  |  |  |  |
| Sapporo 16° | 0.1224 | 0.0034 | 0.0004 |  |  |  |  |  |
| Sapporo 22° | 0.0002 | 0.0063 | 0.0001 | 0.0036 |  |  |  |  |
| Sapporo 28° | 0.0107 | 0.0001 | 0.0055 | 0.0550 | 0.0053 |  |  |  |
| Dayton 16° | 0.0306 | 0.0115 | 0.0001 | 0.0415 | 0.0002 | 0.0002 |  |  |
| Dayton 22° | 0.0107 | 0.3861 | 0.0010 | 0.0790 | 0.0208 | 0.0007 | 0.2093 |  |
| Dayton 28° | 0.0460 | 0.0044 | 0.0060 | 0.1402 | 0.0193 | 0.2199 | 0.0136 | 0.0254 |

variation between individuals ($MS_{IND}/MS_{RES} = 3.92$, $p = 0.011$). Although substantial, measurement error due to three-dimensional reconstruction and landmarking processes should not preclude detection of genuine variation among individual ovipositors.

## 3.2. Temperature and population effects

Overall, both temperature ($Z = 5.27$, $p < 0.001$) and geography ($Z = 4.72$, $p < 0.001$) had a significant effect on ovipositor shape. In addition, temperature interacted with geography in their association with shape ($Z = 1.95$, $p = 0.026$), suggesting a different effect of temperature among geographical populations. The pairwise comparisons between geographical samples showed that the significance of this interaction was driven by a subtle difference between Sapporo and Paris populations ($Z = 1.96$, $p = 0.035$).

The temperature shift from 22 to 16°C is associated with a narrowed ovipositor overall (figure 3). At 16°, the ovipositor seemed to be elongated and flatter, producing an inner folding of the lateral parts of the ovipositor within the structure and therefore smaller and plane lateral parts. The increase from 22 to 28°C produced again an overall narrowing of the ovipositor (although less pronounced than at 16°) and the widening of the anterior part of the ovipositor. In comparison with Sapporo population, Paris population showed a narrower posterior part and more folding on the lateral parts, which were smaller (figure 4). Dayton seemed the most elongated geographical population and the one with the narrowest anterior part.

shape changes in response to temperature

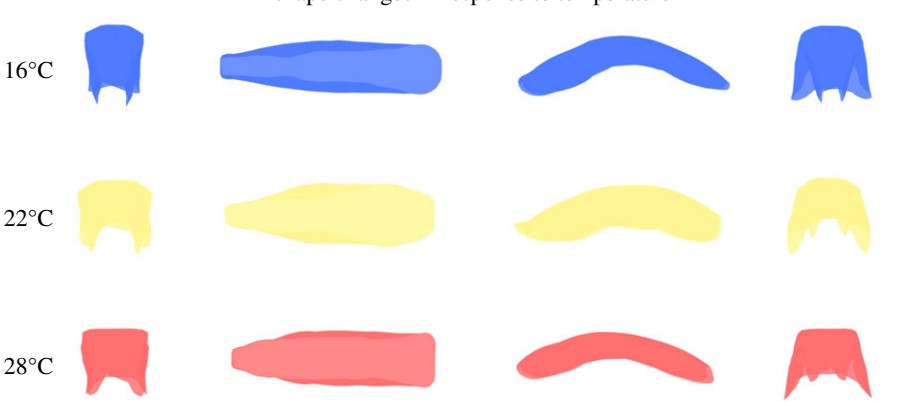

**Figure 3.** Effect of developmental temperature on the ovipositor three-dimensional shape. While the ovipositor shape at 22°C (centre row) represents the approximate real shape of the three populations at that temperature, morphologies at extreme temperatures are represented as exaggerated versions (five standard deviations) of the linear transformation from 22°C to each temperature. Therefore, the linear transformation from 16° to 28°, not biologically meaningful as the effect of temperature is not linear, is not represented. Three-dimensional shapes are captured by four different perspectives (from left to right: posterior, dorsal, lateral and anterior). Shape changes were obtained with the Morpho library. See results for the description of the shape changes.

shape changes associated to geographic variation

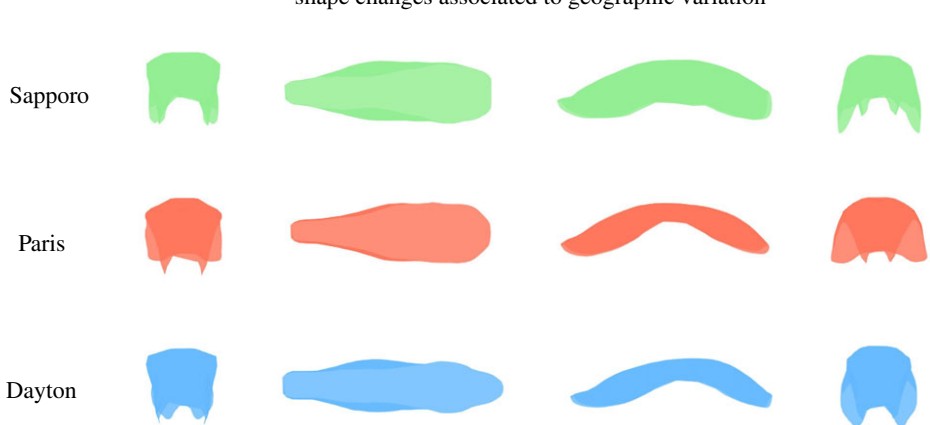

**Figure 4.** Effect of geographical variation on the ovipositor three-dimensional shape. Here, we represent the exaggerated linear deformation (three standard deviations) from the overall mean shape to each geographical population shape. Three-dimensional shapes are captured by four different perspectives (from left to right: distal, dorsal, lateral and proximal). Similar to figure 3, the linear transformation from the Paris to Dayton population does not make biological sense since both come from a Japanese population [12]. Therefore, Sapporo population is represented by its true mean shape and the other two populations as a linear transformation from the former to each of the latter populations. Shape changes were obtained with the Morpho library. See results for the description of the shape changes.

The trajectory analysis showed a striking conservation of the shape variation patterns among geographical populations (figure 2). Trajectories for all geographical populations showed very similar path lengths (Paris = 0.10, Sapporo = 0.08, Dayton = 0.10) and no difference was detected (Sapporo–Paris: effect size = −0.02, $p = 0.41$, Sapporo–Dayton: effect size = −0.48, $p = 0.63$, Paris–Dayton: effect size = −1.05, $p = 0.87$). Although angles among populations showed larger variation, no difference among trajectory angles was found (Sapporo–Paris: angle = 120.56°, effect size = 0.98, $p = 0.977$, Sapporo–Dayton: angle = 100.36°, effect size = 0.48, $p = 0.361$, Dayton–Paris: angle = 41.92°, effect size = −0.96, $p = 0.77$). Similarly, shape differences among trajectories were not significant (Sapporo–Paris: Procrustes distance = 0.10, effect size = −1.16, $p = 0.89$, Sapporo–Dayton: Procrustes distance = 0.25, effect size = −0.04, $p = 0.47$, Paris–Dayton: Procrustes distance = 0.17, effect size = −0.70, $p = 0.74$).

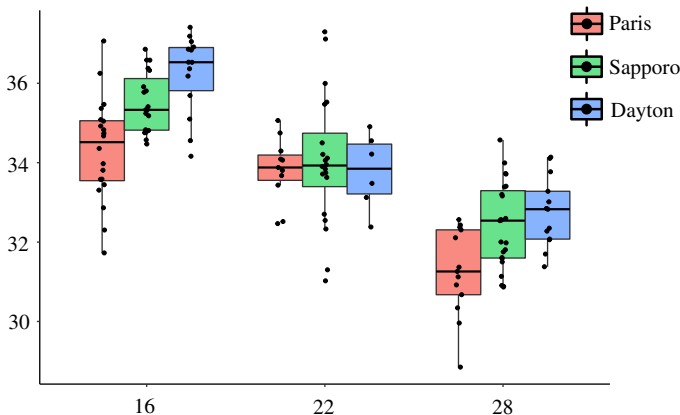

**Figure 5.** Effect of developmental temperature on the ovipositor centroid size. Ovipositor centroid size variation in response to developmental temperature (16°C: left block, 22°C: middle block, 28°C: right block), for each population (Paris: red, Sapporo: green, Dayton: blue).

## 3.3. Size variation and allometry

The ovipositor size was found to decrease with increasing temperature (figure 5, $F_{2,130} = 92.31$, $p < 0.001$). Geography also showed a significant effect on the ovipositor size (figure 5, $F_{2,130} = 14.875$, $p < 0.001$), Dayton populations being larger than Paris. No interaction between temperature and population effects was detected ($F_{4,130} = 2.138$, $p = 0.08$), suggesting that the plasticity of ovipositor size was conserved across populations.

Ovipositor shape and size were correlated, so the plastic response to temperature produced a general allometric pattern ($Z = 3.79$, $p < 0.001$). When the allometric slope among geographical populations was compared, no significant difference was found ($Z = 0.49$, $p = 0.325$).

## 3.4. Comparison with the wing

Wing shape showed much larger Mahalanobis distances among temperatures than the ovipositor shape, suggesting that wing shape is more plastic than ovipositor shape. For the ovipositor, the distances from 22°C to the extreme temperatures are relatively stable: 2.38 to 16°C and 3.03 to 28°C. For the wing, both distances were larger but the high temperature had a stronger impact on shape: 2.87 to 16°C ($p < 0.001$) and 5.60 to 28°C ($p < 0.001$). When we look at the distances between the extreme temperatures, the difference between structures became more evident: we obtained a measure of 3.74 from 16 to 28°C for the ovipositor and a measure of 7.78 for the wing ($p < 0.001$).

For the centroid size, within temperature CV were close to 3% for both the wing and the ovipositor (wing: $CV_{16°C} = 3.07\%$, $CV_{22°C} = 3.86\%$, $CV_{28°C} = 2.19\%$; ovipositor: $CV_{16°C} = 3.67\%$, $CV_{22°C} = 3.86\%$, $CV_{28°C} = 3.67\%$). When comparing CV between structures within each temperature, only significant differences at 28°C were found ($MSLRT_{16°C} = 1.14$, $p = 0.286$; $MSLRT_{22°C} < 0.01$, $p = 0.954$; $MSLRT_{28°C} = 8.05$, $p = 0.005$). The wing showed a much larger plastic response among temperatures than the ovipositor ($CV_{WING} = 14.28\%$, $CV_{OVIPOSITOR} = 4.55\%$, $MSLRT = 65.42$, $p < 0.001$).

## 4. Discussion

Our results showed significant but limited plasticity of the ovipositor shape to developmental temperature in comparison with the wing, suggesting a high robustness of the former structure against environmental variation. We also found some geographical variation associated with the ovipositor shape but its effect seemed subtle as well. This variation probably arises as a consequence of the geographical spread of this species over the last years [12]. Although the interaction between temperature and geographical variation appeared significant, we did not find differences among reaction norms in either trajectory size, direction or shape. The allometry test confirmed these results from a different perspective: developmental temperature produces a particular relationship between the ovipositor size and shape that appeared stable among geographical populations.

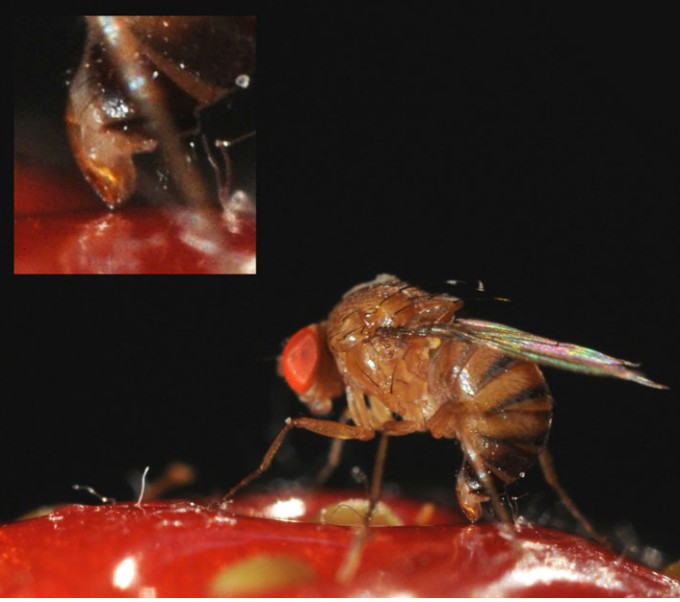

**Figure 6.** *Drosophila suzukii* ovipositing on a strawberry. Copyright: Yann Le Poul.

Developmental temperature is a well-known factor in the origin of size and shape variation in insects [22,47]. In the ovipositor, we found the expected effect of developmental temperature (i.e. higher temperature, smaller ovipositors) [24] and the expected presence of allometry published for two-dimensional analyses [16]. Our three-dimensional approach allowed us to depict and quantify the full shape of the ovipositor and should thus allow detecting any differences among temperature and geographical factors.

In the light of our estimates, and especially if we compare the effect of temperature on the ovipositor size with that on wing size in the same populations under the same experimental design [10], the ovipositor appears to be somewhat robust to temperature. This robustness is consistent with previous studies of phenotypic plasticity in Drosophila, showing a reduced variability of genitalia compared with other body parts [21,48]. The mild plastic variation expressed in our experiments and the success of the invasion suggest that the ovipositor is able to perform well in a wide range of environmental conditions. One possible explanation for the limited plasticity of the ovipositor could be the effect of stabilizing selection, which could reduce its range of variation. This limited plasticity is congruent with the limited geographical variation detected and previous evidence on coevolution of the ovipositor with the male genitalia [49], expected for a trait under stabilizing selection [50,51]. A formal Qst/FSt comparison [52] would nevertheless be necessary to test this hypothesis. Directional selection has also been shown to reduce plasticity [53,54], e.g. as a consequence of the differences in the individual developmental variation in response to temperature [55]. Because the individual is the target of selection and not particular structures within the organism, directional selection favouring certain traits may impact the population variability for a different trait. Depending on the developmental covariation between these two traits, directional selection in one trait may result in a small variation for the second one.

Albeit limited, some plasticity in the ovipositor was nevertheless detected, that might have consequences on the female ability to pierce the fruit tegument. Temperature enhances fruit ripening and this change in the fruit consistency (weakening the surface) might impose new functional demands on the ovipositor morphology to successfully perforate the fruits during the oviposition (figure 6). Although fully hypothetical, it is conceivable that the plastic shape changes reported here might have some adaptive value. This should be tested experimentally by evaluating the relative performance on a variety of substrates, of the cold- and hot-generated ovipositors. Other factors like the existence of alternative selective pressures imposed on the ovipositor morphology such as sexual coevolution [49] and pleiotropic genetic effects during the ovipositor development [56] might limit such morphological adaptation.

The lack of difference in plasticity between invasive and native populations suggests that the role of plasticity in the ovipositor during the worldwide invasion of *D. suzukii*, if any, has been limited. A similar result was found for wing shape plasticity, using males from the same populations [10]. It has been proposed that plasticity might be transient during colonization [3], leaving open the possibility that

plasticity might have contributed to the invasion success prior being genetically fixed. Given the speed of *D. suzukii* invasion [12] and the fact that all three populations show limited plastic responses, such hypothesis of 'rapidly evolving' plasticity nevertheless seems unlikely.

In conclusion, while we detected some genetic divergence among populations and some thermal plasticity, phenotypic variation of the ovipositor was very limited, suggesting a high phenotypic robustness indicative of a history of stabilizing selection. The lack of difference in plasticity among populations suggests that the ovipositor large performance spectrum and phenotypic robustness rather than its plasticity would contribute to *D. suzukii* invasive success.

Data accessibility. All data and materials are available at the Dryad Digital Repository: https://doi.org/10.5061/dryad.m63Xsj3X5. [57]

Authors' contributions. A.F., V.D. and R.C.: conceptualization; A.F. and V.D.: flies collection; A.F.: experimentation and microscopy data collection; C.V.-G. and A.D.: photogrammetric data collection; C.V.-G.: morphometric analysis; C.V.-G., V.D. and R.C.: results, discussion and writing.

Competing interests. The authors declare no competing or financial interests.

Funding. This study was funded by the Agence Nationale de la Recherche (ANR) under the project SWING (grant no. ANR-16-CE02-0015).

Acknowledgements. We would like to thank K. Tamura, M. Toda, P. Shearer, S. Fellous and T. Schlenke for their help during the fly sampling. We also thank M. Guillaume for her help with the fly stock maintenance and F. Peronnet for the rearing medium we used. ESEM imaging was performed at the Plateform MEA, University of Montpellier. We also would like to thank Géraldine Toutirais of the 'Plateau Technique de Microcopie Electronique du MNHN' and Céline Houssin for her technical expertise in electronic microscopy. S. Gerber helped with the interpretation of some of the methods. This article was also improved by the comments made during the SWING project meetings and the different people attending the seminars where it was presented, as the GDR PlasPhen 2018 in Lyon.

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
