## [Reviewer comments · Royal Society Open Science]

Review History

RSOS-191577.R0 (Original submission)

Review form: Reviewer 1

Is the manuscript scientifically sound in its present form?

Yes

Are the interpretations and conclusions justified by the results?

Yes

Is the language acceptable?

Yes

Do you have any ethical concerns with this paper?

No

Have you any concerns about statistical analyses in this paper?

No

Recommendation?

Accept as is

Comments to the Author(s)

I enjoyed reading the manuscript and the flow is coherent and clear. The authors explored whether phenotypic plasticity on the *Drosophila suzukii* ovipositor, a key feature, that allows this species to cause severe damage to fruit crops, plays a role on this pest global expansion. In this case, it seems that robustness allied with a broad capacity of performance is helping the success of this pest species. This assessment will contribute to our knowledge of the role of plasticity on biological invasions and new niche colonizations.

I also want to add that I particularly appreciated the combination of SEM and phtogrammetry techniques and geometric morphometrics to address the 3dimensionality of the ovipositor structure.

Typos/small mistakes:

Fig. 4 (Legend) - it is written "effect of developmental temperature", when in reality it should be "geographic location"

Review form: Reviewer 2

Is the manuscript scientifically sound in its present form?

Yes

Are the interpretations and conclusions justified by the results?

Yes

Is the language acceptable?

Yes

Do you have any ethical concerns with this paper?

No

Have you any concerns about statistical analyses in this paper?

No

Recommendation?

Reject

Comments to the Author(s)

The manuscript of Varón-González and colleagues is a very well conceived study that investigates the phenotypic plasticity of *Drosophila suzukii* ovipositor at different temperatures in population from different origin. The manuscript is well written and the techniques adopted for the geometric morphometry are robust and well suited to investigate the subject. Said that, the study is primarily confirmative of already published results and its novelty is limited to the comparison among three disjoint populations and the finding that the ovipositor structures remain highly stable across a large geographical range. Maybe, a more large sampling of natural populations to better cover the species distribution, than with just three samples, would have make this study more attractive.

While I may understand the need to build a story on a running hypothesis I cannot find the one proposed by authors as a reliable one. Indeed, affirming that "It is thus conceivable that *D. suzukii* ovipositor might present some adaptive plasticity to temperature, allowing it to pierce fruits skins of (thermally induced) varying resistance" (page 3, lines 81-83) implies that the ovipositor structures may undergo to relaxation when the species face soft fruit skin or

conversely gain robustness when the populations ovoposit on hard skin fruit. However, *D. suzukii* is a highly polyphagous species that use small fruits of a large array of crop and non-crop species (Burrack, H. J. 2013. Variation in selection and utilization of host crops in the field and laboratory by *Drosophila suzukii* Matsumara (Diptera: Drosophilidae), an invasive frugivore. *Pest Manag Sci* 69(10): 1173; Kenis, M., Tonina, L., Eschen, R., van der Sluis, B., Sancassani, M., Mori, N., ... & Helsen, H. (2016). Non-crop plants used as hosts by *Drosophila suzukii* in Europe. *Journal of Pest Science*, 1-14; Tonina, L., Mori, N., Giomi, F., & Battisti, A. (2016). Development of *Drosophila suzukii* at low temperatures in mountain areas. *Journal of Pest Science*, 1-12), thus it encounters a large variety of skins texture and hardness within the very same generation. On this premises, it is hardly conceivable that a weaker form of ovopositor may be selected in warmer climate. Thus, the hypothesis should be based on a more robust and likely theoretical background.

The titles of figure 3 and 4 are the same and this is a bit misleading.

Decision letter (RSOS-191577.R0)

12-Nov-2019

Dear Dr Varón-González

On behalf of the Editors, I am pleased to inform you that your Manuscript RSOS-191577 entitled "Limited thermal plasticity and geographic divergence in the ovipositor of *Drosophila suzukii*" has been accepted for publication in Royal Society Open Science subject to minor revision in accordance with the referee suggestions. Please find the referees' comments at the end of this email.

The reviewers and handling editors have recommended publication, but also suggest some minor revisions to your manuscript. Therefore, I invite you to respond to the comments and revise your manuscript.

- Ethics statement

- Data accessibility

If you wish to submit your supporting data or code to Dryad (<http://datadryad.org/>), or modify your current submission to dryad, please use the following link:
<http://datadryad.org/submit?journalID=RSOS&manu=RSOS-191577>

- **Competing interests**

- **Authors' contributions**

- **Acknowledgements**

- **Funding statement**

Because the schedule for publication is very tight, it is a condition of publication that you submit the revised version of your manuscript before 21-Nov-2019. Please note that the revision deadline will expire at 00.00am on this date. If you do not think you will be able to meet this date please let me know immediately.

If your manuscript is newly submitted and subsequently accepted for publication, you will be asked to pay the article processing charge, unless you request a waiver and this is approved by Royal Society Publishing. You can find out more about the charges at <https://royalsocietypublishing.org/rsos/charges>. Should you have any queries, please contact openscience@royalsociety.org.

Kind regards,
Lianne Parkhouse
Editorial Coordinator
Royal Society Open Science
openscience@royalsociety.org

on behalf of Dr Richard Benton (Associate Editor) and Kevin Padian (Subject Editor)
openscience@royalsociety.org

Editors' comments to Author:

Thanks for your submission. One reviewer is happy with it as is and the other seems to think it does not show much that is new. However the study of geographical variation in the ovipositor is

useful and I hope you will consider that reviewer's suggestions for the literature relevant to the question. Best wishes for your revision.

Reviewer comments to Author:

Reviewer: 1

Comments to the Author(s)

I enjoyed reading the manuscript and the flow is coherent and clear. The authors explored whether phenotypic plasticity on the *Drosophila suzukii* ovipositor, a key feature, that allows this species to cause severe damage to fruit crops, plays a role on this pest global expansion. In this case, it seems that robustness allied with a broad capacity of performance is helping the success of this pest species. This assessment will contribute to our knowledge of the role of plasticity on biological invasions and new niche colonizations.

I also want to add that I particularly appreciated the combination of SEM and photogrammetry techniques and geometric morphometrics to address the 3dimensionality of the ovipositor structure.

Typos/small mistakes:

Fig. 4 (Legend) - it is written "effect of developmental temperature", when in reality it should be "geographic location"

Reviewer: 2

Comments to the Author(s)

The manuscript of Varón-González and colleagues is a very well conceived study that investigates the phenotypic plasticity of *Drosophila suzukii* ovipositor at different temperatures in population from different origin. The manuscript is well written and the techniques adopted for the geometric morphometry are robust and well suited to investigate the subject. Said that, the study is primarily confirmative of already published results and its novelty is limited to the comparison among three disjoint populations and the finding that the ovipositor structures remain highly stable across a large geographical range. Maybe, a more large sampling of natural populations to better cover the species distribution, than with just three samples, would have make this study more attractive.

While I may understand the need to build a story on a running hypothesis I cannot find the one proposed by authors as a reliable one. Indeed, affirming that "It is thus conceivable that *D. suzukii* ovipositor might present some adaptive plasticity to temperature, allowing it to pierce fruits skins of (thermally induced) varying resistance" (page 3, lines 81-83) implies that the ovipositor structures may undergo to relaxation when the species face soft fruit skin or conversely gain robustness when the populations ovoposit on hard skin fruit. However, *D. suzukii* is a highly polyphagous species that use small fruits of a large array of crop and non-crop species (Burrack, H. J. 2013. Variation in selection and utilization of host crops in the field and laboratory by *Drosophila suzukii* Matsumara (Diptera: Drosophilidae), an invasive frugivore. *Pest Manag Sci* 69(10): 1173; Kenis, M., Tonina, L., Eschen, R., van der Sluis, B., Sancassani, M., Mori, N., ... & Helsen, H. (2016). Non-crop plants used as hosts by *Drosophila suzukii* in Europe. *Journal of Pest Science*, 1-14; Tonina, L., Mori, N., Giomi, F., & Battisti, A. (2016). Development of *Drosophila suzukii* at low temperatures in mountain areas. *Journal of Pest Science*, 1-12), thus it encounters a large variety of skins texture and hardness within the very same generation. On this premises, it is hardly conceivable that a weaker form of ovopistor may be selected in warmer climate. Thus, the hypothesis should be based on a more robust and likely theoretical background.

The titles of figure 3 and 4 are the same and this is a bit misleading.

Author's Response to Decision Letter for (RSOS-191577.R0)

See Appendix A.

Decision letter (RSOS-191577.R1)

27-Nov-2019

Dear Dr Varón-González,

It is a pleasure to accept your manuscript entitled "Limited thermal plasticity and geographic divergence in the ovipositor of *Drosophila sukuzii*" in its current form for publication in Royal Society Open Science.

on behalf of Dr Richard Benton (Associate Editor) and Professor Kevin Padian (Subject Editor)
openscience@royalsociety.org

Appendix A

Dear Editor,

We would like to start by thanking you, the associate editor and the reviewers for your work on our manuscript. We appreciate their comments and suggestions, which we hope we have sufficiently addressed in this revision.

First, we have changed the title of the figure 4, which was erroneous as both reviewers noticed. We have now adopted the correction suggested by reviewer 1.

Second, we do not share the opinion expressed by reviewer 2 on the possible adaptive value of the (limited) plasticity we describe in the manuscript, in relation to fruits texture and temperature. *Drosophila suzukii* populations certainly ‘encounter a large variety of skins texture and hardness’ as put by the reviewer but, as shown in the first citation (Burrack 2013), not all fruits are equally affected. It is conceivable that producing a very stiff ovipositor is developmentally costly. The hypothesis of an adaptive developmental adjustment of the ovipositor depending on the temperature (and its associated impact on fruit firmness) seems plausible, although there is no data available to formally assess it. Whether this hypothesis is sufficiently robust or not to be considered is subjective and we think it constitutes an interesting possibility, worth mentioning in the paper.

Given its limited influence on the manuscript (only briefly mentioned in the introduction), we prefer to leave that sentence as it is.

Kind regards,

The authors.